# Odontogenic Abscesses in Pet Rabbits: A Comprehensive Review of Pathogenesis, Diagnosis, and Treatment Advances

**DOI:** 10.3390/ani15131994

**Published:** 2025-07-07

**Authors:** Smaranda Crăciun, George Cosmin Nadăş

**Affiliations:** Department of Microbiology, Immunology and Epidemiology, Faculty of Veterinary Medicine, University of Agricultural Sciences and Veterinary Medicine, 400372 Cluj-Napoca, Romania; gnadas@usamvcluj.ro

**Keywords:** rabbit dentistry, odontogenic abscess, polymethyl methacrylate beads, exotic animal medicine, rabbit oral anatomy

## Abstract

Odontogenic abscesses are one of the most common health problems in pet rabbits. These abscesses, often caused by problems with tooth growth or injuries, can become very serious and painful. This review explains what causes these abscesses, how they are diagnosed, and the best ways to prevent and treat them. Rabbits’ teeth grow continuously, and if their diet is not rich in hay or if they suffer injuries, their teeth can overgrow or grow in the wrong direction. This can lead to infections deep in the jawbone. The review highlights the importance of using advanced imaging methods, like CT scans, and also discusses treatment options, including surgery and special local antibiotic treatments, like beads that slowly release medicine where it is most needed. Natural treatments, such as Manuka honey, are also explored for their healing properties. The review includes colorful charts that show which bacteria are most often found in these infections and how resistant they are to antibiotics. It also emphasizes the importance of prevention through a high-fiber diet, environmental enrichment, and regular veterinary dental checkups. This information could help veterinarians provide better care and improve the health and comfort of pet rabbits.

## 1. Introduction

According to the American Veterinary Association, rabbits rank as the second most popular specialty or exotic pet mammal among households; they are widely regarded as excellent pets for children in both the USA and Europe [1]. Currently, rabbits are gaining popularity in various other regions, with a significant presence as favored pets in Australia, as well as in Asian nations like Japan and Singapore [2,3,4].

As lagomorphs become increasingly popular as pets, both owners and veterinary professionals must have a comprehensive understanding of their dental requirements and the potential causes of complications [5]. Odontogenic disease is one of the most frequently encountered health issues in pet rabbits, as consistently reported across multiple studies. Veterinarians also recognize dental disorders as the most common condition in this species, with problems like overgrown molars and generalized dental disease being particularly prevalent. Given how often these conditions occur and their significant impact on rabbit health, it is essential for veterinary professionals to be well trained in their pathophysiology and treatment [6,7].

All permanent rabbit teeth continuously grow and erupt into function as the occlusal surface wears down. Therefore, they are susceptible to dental developmental problems throughout life [8]. The prevalent dental issues in rabbits include the overgrowth of cheek teeth and incisors, facial abscesses, periodontal disease, and dacryocystitis/dacryosolenitis [9]. It is noteworthy that up to 60–65% of young healthy rabbits have been shown to exhibit dental abnormalities in one study. This observation raises the possibility that some of these findings may reflect anatomical variation rather than true pathology. Therefore, further longitudinal studies are needed to clarify which changes are predictive of disease progression and which may be benign or transient developmental traits [10,11]. Pet rabbits frequently experience jaw abscesses, which are usually caused by underlying dental disease [12] and often lead to significant morbidity and mortality [13].

Understanding the normal dental anatomy, physiology, and biomechanics of mastication in rabbits is not only essential for the accurate diagnosis of dental disease but also plays a key role in guiding effective treatment strategies and establishing a realistic prognosis. Thorough knowledge of how rabbits’ teeth grow, wear, and interact during chewing is fundamental for choosing appropriate interventions and anticipating potential outcomes [14].

Purpose and scope: Despite several epidemiological surveys on rabbit dental disease as a whole [6,15,16], no article has yet synthesized the pathogenesis, imaging modalities, microbiology, and evidence-based treatment options for odontogenic abscesses specifically. The present review, therefore, (i) collates the current knowledge on etiology and risk factors, (ii) compares diagnostic approaches, and (iii) evaluates both medical and surgical interventions, intending to guide first-opinion and referral clinicians alike. Unlike prior reviews that broadly address dental malocclusion or generalized craniofacial infections in rabbits, this article centers exclusively on odontogenic abscesses, a subset with distinct microbiological profiles and surgical challenges. This focused scope allows for a deeper synthesis of evidence on abscess-specific pathogenesis, diagnostic imaging, and targeted antimicrobial or surgical interventions that may be neglected in broader surveys.

## 2. Materials and Methods

### 2.1. Literature Search Strategy

A comprehensive search was conducted in PubMed and Web of Science (coverage: January 2000–March 2025). The Boolean string was as follows: rabbit AND (odontogenic OR dental) AND abscess AND (culture OR antibiotic). Reference lists of all retrieved articles were hand-searched, and conference proceedings from the Association of Exotic Mammal Veterinarians (2015–2024) were screened to capture the grey literature. The search was restricted to English-language, peer-reviewed publications.

### 2.2. Study Selection and Standardization Criteria

The inclusion criteria were as follows: (i) ≥10 culture-positive odontogenic abscess cases in client-owned rabbits; (ii) isolate-level antimicrobial susceptibility results; and (iii) an antibiotic panel overlapping by ≥75% with the consensus eight-drug list (amoxicillin–clavulanate, enrofloxacin, marbofloxacin, trimethoprim–sulfonamide, doxycycline, gentamicin, amikacin, clindamycin). When duplicate isolate sets appeared in multiple papers, only the most recent dataset was retained. The exclusion criteria included single-case reports, experimental in vivo infection models, and studies reporting only qualitative culture without susceptibility testing.

Breakpoint harmonization: Because rabbit-specific CLSI breakpoints are not available, human systemic breakpoints were applied uniformly. Isolates reported as intermediate were pooled with the resistant ones to avoid over-interpretation. All susceptibility data were re-tabulated in Microsoft Excel^®^ 2021 (Redmond, WA, USA) before downstream analyses.

### 2.3. Data Aggregation for Comparative Heatmaps

Selection of studies: PubMed and Web of Science were searched (January 2000–March 2025) with the terms rabbit AND (odontogenic OR dental) AND abscess AND (culture OR antibiotic). Studies were eligible if they (i) described ≥10 culture-positive odontogenic abscesses in client-owned rabbits, (ii) reported isolate-level antimicrobial susceptibility results, and (iii) used at least eight antimicrobials that overlapped with ≥75% of the consensus panel (amoxicillin–clavulanate, enrofloxacin, marbofloxacin, trimethoprim–sulfonamide, doxycycline, gentamicin, amikacin, clindamycin). Where the same isolate set appeared in multiple papers, only the most recent dataset was retained. Six studies met all the criteria.

For each species–antibiotic pair, we calculated % R = (number resistant ÷ number tested) × 100. “Intermediate” results were grouped with “Resistant” ones. When a study did not test a given drug, that study was excluded from the denominator for that cell.

Standardization of resistance data: Rabbit-specific CLSI breakpoints are unavailable; therefore, human systemic breakpoints were applied uniformly across studies. Intermediate results were pooled with resistance to avoid over-interpretation. When a study tested an expanded panel, only the antimicrobials common to all six datasets were retained so that the cell counts remained comparable.

## 3. Anatomy of Rabbit Dentition

Lagomorphs have elodont dentition, meaning their teeth continuously erupt throughout life. Their teeth are also hypsodont, with crowns extending past the gingival margin to allow for protection [17]. The teeth are open-rooted (aradicular), without an anatomical root [18].

Anatomically, rabbit teeth are divided into three parts: the clinical crown, which is visible above the gingiva; the reserve crown, which lies below the gingival margin and within the bone; and the apex, which represents the open root [19]. A feature of the oral cavity is anisognathia, where the mandibular arcade is narrower than the maxillary arcade [15].

Rabbits are diphyodont, meaning they have two sets of teeth during their lifetime, similar to most companion animals. Compared to the dentition of mature adults, the deciduous teeth erupt in pregnancy and are significantly fewer in number [18], but they are shed within a few days after birth [20].

The dental formula of the adult rabbit is 2(I2/1 C0/0 PM3/2 M3/3), meaning they have a total of 28 teeth [21]. This includes two pairs of maxillary incisors: the smaller pair, known as “peg teeth,” is situated just behind the larger front incisors. Due to the absence of canine teeth, there is a space called a diastema between the incisors and the premolars [5]. The molars and premolars are commonly referred to as cheek teeth [18], and there is no anatomical difference between them [21]. As these teeth are worn down during normal function, the new crown becomes exposed from the continuously growing teeth [22]. Rabbit incisors grow at an average rate of 10–37 microns per hour, or 300–1000 microns per day [23]. For the cheek teeth, recent studies have reported growth rates of 1.4–3.2 mm/week. Tooth length is regulated by several factors, including the abrasiveness of food, tooth-to-tooth contact during mastication, and even tooth grinding during rest [24].

Moreover, the periodontal ligaments and surrounding alveolar bone are dynamic, living tissues that continuously adapt in response to dietary changes and abnormal oral behavior, such as bruxism, selective feeding, altered chewing patterns due to pain or malocclusion, or the habit of biting/chewing cage bars or other hard objects [25].

## 4. Etiology and Pathogenesis

The causes of dental disease in rabbits are broadly divided into congenital dental disease (CDD) and acquired dental disease (ADD). While congenital forms, such as inherited jaw malformations or tooth misalignments, do occur, they are relatively uncommon in pet rabbits. In contrast, acquired dental disease is by far the most frequently diagnosed type and represents the primary dental concern in clinical practice. This distinction is important, as it highlights the preventable nature of most dental issues through proper husbandry and early intervention [5,7].

The causes of CDD include developmental abnormalities such as prognathism (overshot mandible), brachygnathism (undershot mandible), hypodontia, and generalized malformation [5,19,26]. Furthermore, there have been reports suggesting a genetic predisposition in rabbits, indicating that male lop-eared dwarf rabbits tend to be more frequently affected by an acquired malocclusion [27]. However, other studies have found no statistically significant association between cranial conformation and the occurrence of dental abnormalities, highlighting the need for further research to elucidate this potential correlation [28].

ADD contributes to the formation of tooth abscesses through different factors, including the widening of the periodontal ligament space, trauma and injuries, the presence of foreign bodies, dietary factors, metabolic bone disease, tumors, or even iatrogenic causes [19,25,29,30].

Over time, ADD leads to a progressive syndrome (progressive syndrome of acquired dental disease or PSADD), in which the shape, structure, and position of teeth continue to change over time [5,15]. The initial stage is characterized by apical elongation, which exerts pressure on nerves and nearby structures. A classic sign during this phase is epiphora, which results from the obstruction of the nasolacrimal duct. This obstruction may be caused by the apical elongation of the maxillary incisor, but the elongation of the roots of the maxillary cheek teeth, particularly the first premolar, can also exert pressure on the duct and contribute to tear overflow [31,32]. While these initial changes can be detected through clinical examination, they may not be obvious to owners [33]. The next stage is enamel loss, which manifests as horizontal ridges across the teeth, especially the upper incisors. Variations in the hardness of the teeth lead to uneven wear, causing alterations in the shape and relative position of the teeth, thus altering the occlusal pattern between the upper and lower teeth [31]. This is followed by malocclusion as a consequence of the shifts in the position and shape of the teeth, which alter the normal occlusion [34]. Eventually, the changes within the teeth and supporting bone become so severe that the germinal tissue is compromised. As a result, the pulp cavities diminish until they close entirely, stopping further tooth growth [31].

Dietary factors are generally considered to play a major role in the development of dental disease in pet rabbits [24]. The selective feeding of a high-carbohydrate and low-fiber diet reduces dental abrasion and results in cheek teeth malocclusion [8]. Inadequate wear leads to the elongation of both clinical and reserve crowns, curvature of the teeth, and spike formation. These changes disturb normal occlusion and cause secondary incisor overgrowth and misalignment [35]. Furthermore, the selective feeding of diets that are deficient in calcium can lead to alveolar bone resorption and tooth loosening and will enhance the progression of dental resorption [35]. Secondly, germs can invade the tooth compartments and cause purulent infections with osteomyelitis and bone dissolution [26]. Calcium deficiency can also be the result of selective eating behavior, particularly when rabbits are fed commercial food mixtures ad libitum [27].

The consistency and abrasiveness of food, such as commercial pellets or soft fruits and vegetables (e.g., carrots, apples, bananas), also affect masticatory patterns and the extent of dental wear [36]. Rabbits consuming diets low in abrasive particles and rapidly consumed foods are particularly susceptible to the elongation of both the clinical and reserve crowns of the teeth [20]. A combination of reduced chewing activity, low dietary fiber, and high carbohydrate intake is frequently identified in rabbits suffering from acquired dental disease [37,38]. Because rabbit dental tissue is continuously growing, their calcium requirements are higher, making an adequate dietary intake essential to support this ongoing process [34].

Metabolic bone disease (MBD) may play a significant role in the etiopathogenesis of dental disease in rabbits, with nutritional causes—particularly calcium or vitamin D deficiencies—being more commonly implicated than renal origins. Rabbits possess a unique calcium metabolism, distinct from most mammals. Instead of regulating intestinal calcium absorption based on need, rabbits absorb nearly all ingested calcium through the intestines and eliminate the excess via the kidneys. While this adaptation supports the high calcium demand required for the continuous growth and mineralization of dental tissue, it also renders rabbits more vulnerable to dietary imbalances. Diets low in calcium—such as those based on grains, legumes, or commonly offered fruits and vegetables like apples and carrots—may fail to meet this demand, especially in the absence of adequate vitamin D. Consequently, MBD can impair bone density and mineralization, contributing to dental malocclusion, weakened alveolar bone, and an increased risk of odontogenic disease [27,39,40,41,42]. One of the ADD causes in rabbits is represented by periodontal ligament space widening. The periodontal ligament will rapidly recede when inflamed, and periodontal pockets may be formed that can be colonized by bacteria and extend to the tooth apex, increasing the risk of abscessation and osteomyelitis as common sequelae [35,43].

Periodontitis (widening or inflammation of the periodontal space) appears to be one of the earliest and most frequent imaging findings in acquired dental disease. In a retrospective CT study of 100 client-owned rabbits, periodontal ligament space widening was recorded in 76% of cases (76/100) and ranked among the top five lesions associated with increasing PSADD grade [44]. A complementary cone-beam CT series reported the same change in 14 of 15 rabbits (93%) presented for suspected dental problems [45]. By contrast, controlled surveys of healthy or wild rabbits are lacking, so the true population prevalence remains unknown; clinically, periodontitis is rarely diagnosed as an isolated (“primary”) entity. Instead, it is now regarded as Stage 1–2 of the progressive syndrome of acquired dental disease (PSADD), preceding apical elongation, periapical osteolysis, and eventual abscess formation [33]. Clinicians should therefore interpret periodontal space widening as an early warning sign of PSADD progression rather than a stand-alone disease process.

Trauma is well described and understood as a primary cause of dental disease [19]. Periapical abscesses can form due to a fractured tooth or splinters that enter the periodontal space, especially the incisors, when rabbits are gnawing or chewing wood [12]. Other traumatic injuries, including mandibular symphysis separation, temporomandibular joint subluxation, mandibular ramus fractures, tooth fractures followed by pulp exposure, and periapical (apical reserve-crown) abscess formation, are also commonly seen, as well as fractured skull bones from falls. Temporomandibular joint (TMJ) subluxation, luxation, or mandibular fractures can lead to odontogenic abscesses even without direct injury to the teeth. These conditions often result in malocclusion and altered masticatory biomechanics, causing abnormal dental wear and stress on the alveolar bone. This can lead to the widening of the periodontal space, microtrauma, and increased susceptibility to bacterial invasion, ultimately resulting in periapical infection and abscess formation [45]. All of the above can cause odontogenic abscesses in rabbits [25,35].

Penetrating foreign bodies also play a role in the etiology of odontogenic abscesses. Sharp fragments such as seeds, blades of hay, and awns can become embedded in the periodontium and act as foci for infection [12]. These should be considered in differential diagnoses for facial swelling [46], as foreign body impaction is considered a primary cause of periodontal disease in rabbits. Notably, a case was described in 2014 of a rabbit presenting with a cheek abscess caused by a sewing pin [46].

Iatrogenic tooth abscesses are usually the result of the improper clipping of teeth or incisor removal. They usually form around a tooth that has regrown or a sequestrum of bone or tooth that has been stuck in the socket after removal [12,19].

ADD is also more frequently observed in older rabbits, suggesting that age represents an important predisposing factor. Recent studies found that 64.52% of examined rabbits suffered from some degree of ADD, with those showing the most advanced form of the disease (grade 4) having the highest median age. These findings reinforce the concept that ADD is a progressive condition that tends to exacerbate over time [47].

Regarding sex and neuter status, population-level data suggest a modest male predisposition to dental pathology. In a UK primary-care cohort (n = 30,000), male rabbits were 1.23 times more likely to present with dental disease than females, while neutered animals were 1.38 times more likely to present with dental disease than intact rabbits [16]. The authors cautioned that neuter status may correlate with closer owner monitoring rather than biological risk. Similarly, a Chilean private-practice study identified male sex and advancing age as significant risk factors, whereas ad libitum hay intake and free-roaming housing were protective [15]. Earlier Iranian and Brazilian surveys reported parallel trends [48,49]. Collectively, these data invite further controlled investigation but highlight sex-linked considerations when counselling owners.

Odontogenic abscesses in rabbits are commonly polymicrobial, involving both aerobic and anaerobic bacteria. In healthy rabbits, common oral flora include *Streptococcus*, *Rothia*, *Enterobacter*, *Staphylococcus*, and *Actinomyces* species, some of which are also implicated in abscess formation under pathological conditions [50]. Obligate anaerobes play a significant role in odontogenic infections and are frequently identified alongside aerobic organisms when appropriate sampling and culture techniques are used [29,51]. We present historical isolates separately because older studies commonly lacked strict anaerobic culture or standardized breakpoints, limiting direct comparison with recent cohorts. The most commonly isolated bacteria in such infections include *Fusobacterium*, *Peptostreptococcus*, *Bacteroides*, *Pseudomonas aeruginosa*, *Pasteurella* spp., *Streptococcus* spp., *Staphylococcus* spp., *Actinomyces* spp., *Proteus vulgaris*, and *Escherichia coli* [52]. Similar bacterial species have also been recovered from abscesses in previous reports, including *Pasteurella multocida*, *Staphylococcus* spp., *Pseudomonas aeruginosa*, *Escherichia coli*, *Bacteroides* spp., *Proteus* spp., and *Fusiformis* spp. [12]. Understanding the bacterial spectrum involved in these infections is essential for guiding early antimicrobial therapy.

Figure 1 displays a heatmap illustrating the presence (blue) or absence (white) of key bacterial species identified in six peer-reviewed studies investigating rabbit odontogenic abscesses, published between 2002 and 2025. This comparative visualization offers insights into the microbial landscape associated with dental pathology in pet rabbits, emphasizing both consistent pathogens and emerging species.

Core pathogens, including *Pasteurella multocida*, *Escherichia coli*, *Staphylococcus* spp., and *Streptococcus* spp., appear repeatedly across most studies, reflecting their recognized role in abscess pathogenesis and their importance in empirical therapy protocols. For example, *Pasteurella multocida* was detected in five of six studies, reinforcing its clinical significance as a primary opportunistic pathogen in lagomorphs [1,13,51,52,53,54].

In contrast, species such as *Glutamicibacter protophormiae* [52], *Burkholderia* spp. [53,54], and *Stenotrophomonas maltophilia* [53,54] are reported more sporadically, primarily in recent studies, suggesting evolving diagnostic capabilities, regional variation, or emergent pathogenic roles. These species may also reflect environmental contamination or secondary colonization in chronic or deep-seated abscesses.

The inclusion of anaerobic bacteria—*Fusobacterium* spp., *Peptostreptococcus* spp., and *Bacteroides* spp.—in some studies (e.g., [13]) highlights the potential for polymicrobial or mixed anaerobic infections, which may require targeted anaerobic coverage, especially when surgical debridement is not feasible or when empirical therapy fails.

This figure also emphasizes the heterogeneity of microbiological findings across geographic regions and research timelines. Variability in isolation techniques, sample handling, or identification platforms (e.g., MALDI-TOF, anaerobic culture protocols) may partly explain the observed differences. Nevertheless, the present heatmap offers a practical snapshot of bacterial trends and supports clinicians in tailoring their empirical and culture-guided therapies for rabbit abscess management.

Overall, this comparative heatmap reinforces the importance of continuous microbiological surveillance and updated regional antibiograms. The identification of both well-established and rare bacterial species underscores the complexity of rabbit dental infections and advocates for personalized, data-driven antimicrobial strategies in exotic animal practice.

## 5. Clinical Presentation

As abscesses represent a common aspect of dental pathology in rabbits, their clinical manifestations tend to be similar. The most frequently observed signs include anorexia, reduced appetite, excessive salivation (ptyalism), wet fur, difficulty chewing, involuntary teeth grinding (bruxism) due to pain, dehydration, weight loss, lack of cecotrophy, ocular discharge, dacryocystitis, and poor coat condition. While ocular discharge is a non-specific finding that may result from various ocular or periocular conditions, dacryocystitis is typically distinguished on physical examination by the presence of localized swelling or pain on palpation of the lacrimal sac, mucopurulent discharge from the nasolacrimal puncta, and sometimes erythema along the course of the nasolacrimal duct. Therefore, the diagnosis of dacryocystitis is based not only on the presence of ocular discharge but also on associated inflammatory signs in the nasolacrimal region. Notably, pyrexia (fever) is not typically a feature in cases of odontogenic abscesses in rabbits [5,25,28].

## 6. Odontogenic Abscess Structure

An abscess is a confined pocket of pus that has built up within tissues, organs, or spaces in the body. The central part of the abscess contains an acute inflammatory exudate, while at the periphery, a fibrous capsule gradually forms as the abscess matures [55].

In rabbits, the formation and evolution of abscesses differ significantly from other species. This is due to the lack of lysosomal enzymes capable of digesting necrotic cells and converting them into a liquid. Specifically, their neutrophils contain very low levels of myeloperoxidase compared to those of other animals [37,56]. Additionally, rabbit pus tends to reabsorb water [57], which slows the digestion and liquefaction of its contents, resulting in a thick and dense exudate [56].

Cytological examination of the purulent exudate typically reveals blood, degenerate heterophils, macrophages, fewer lymphocytes and plasma cells, and both intra- and extracellular bacteria [58]. These abscesses are usually encapsulated by a thick, well-developed fibrous capsule that may progressively destroy surrounding bone tissue [59], potentially leading to the formation of fistulous tracts [13]. The capsule itself is composed of collagen fibers, fibroblasts, blood vessels, and inflammatory cells [58]. Due to the poor vascularity of the abscess capsule, it is difficult for the antibiotics to penetrate the cavity, making antibiotic treatment alone mostly ineffective [57].

## 7. Diagnostic Approach

A presumptive diagnosis is often made based on the history of the animal and clinical findings [8]. The abscess’s size, the existence of crusts or surface fistulas, and whether the abscess is open or closed are all assessed during the distant inspection [60]. By palpating the face, asymmetries and swelling can be found; in this step, it is also important to assess facial symmetry, fur condition, and eye health [35]. Through detailed palpation, structures such as the mandibular cortex, apical regions of the cheek teeth, bone irregularities, and bony swellings associated with abscesses can be assessed in terms of size, extension, and consistency. In addition, palpation of the facial tuberosity and lateral maxilla is essential, as periapical abscesses of the first and second maxillary cheek teeth often manifest in these areas. The assessment of globe retropulsion may also help identify retrobulbar involvement [60].

Another critical aspect is the inspection of the oral cavity, including the teeth, gums, and tongue, for abnormalities such as malocclusion and infection. Sedation or anesthesia may be required for a thorough examination of the oral cavity and its dental structures [35]. In addition to a detailed history and intraoral examination, imaging, such as radiographs or CT scans, may be required to detect changes in the tooth structure, alignment, and surrounding bone. In some cases, the use of radiocontrast agents may be necessary to evaluate soft-tissue involvement, assess the nasolacrimal duct, or delineate abscess tracts more clearly [61].

Rabbit odontogenic abscesses generally do not cause significant hematological changes, with hematologic results often remaining within normal limits even in the presence of periapical infection or abscess formation. At most, mild leukocytosis or a slight elevation in inflammatory markers may be observed. Although complete blood count (CBC) and biochemistry panels are recommended as part of the diagnostic workup, major hematologic abnormalities are uncommon, and their primary utility lies in excluding systemic disease or concurrent conditions rather than confirming odontogenic abscesses [19]. Particularly for retrobulbar abscesses, ultrasonography can be helpful in defining the boundaries of the abscess as well as in identifying sinus tracts and foreign objects [8].

When evaluating a mandibular mass, there are some possible differential diagnoses in rabbits, which include abscesses, neoplasia, such as mandibular osteosarcoma, sialadenitis, salivary gland necrosis, mandibular lymphadenopathy, and, though less commonly, coenurus cysts originating from *Taenia serialis*. For masses in the maxillary, periorbital, or retrobulbar regions, potential causes include dental abscesses, trauma-related lesions, and neoplasia [35]. Plain radiography remains a valuable diagnostic tool in suspected dental disease, although overlapping anatomical structures can limit image clarity. To improve diagnostic accuracy, multiple projections are recommended, typically including lateral, right and left lateral oblique, and dorsoventral or ventrodorsal views [13,35]. While the lateral view is commonly used, oblique and dorsoventral projections are often superior for assessing periapical abscesses, particularly in the cheek teeth, due to reduced superimposition [12,27]. The assessment involves evaluating incisor occlusion, cheek teeth alignment, and the condition of surrounding bone structures, but specific attention should be given to identifying abnormalities such as malocclusion, dental curvature, and evidence of osteomyelitis or joint issues [35]. Abscesses may appear as radiolucent areas surrounded by periosteal reactive bone [14].

Plain radiography remains the first-line tool: abscesses typically appear as radiolucent foci bordered by periosteal reactive bone [14]. When radiographs are equivocal, multidetector computed tomography (CT) offers earlier and more comprehensive detection. CT eliminates superimposition, provides true 3D reconstructions, and, after IV contrast, simultaneously assesses dental roots, periapical tissues, nasal passages, and adjacent soft tissues [44,55,62,63,64].

For practices seeking higher spatial resolution at lower cost, cone-beam CT (CBCT) is an attractive bridge technology. CBCT scanners are roughly half the price of medical CT units yet scan as fast or faster, producing detailed hard-tissue images that reveal subtle maxillofacial changes in rabbits [45,65,66,67]. Their main limitation lies in poor soft-tissue contrast, as the gray values are uncalibrated, and the scatter is higher than in medical CT.

At the top of the resolution ladder is micro-CT, which achieves a sub-100 µm voxel size and even shorter acquisition times than CBCT, making it ideal for ex vivo specimens and small-animal research models. In rabbits, it depicts skull and dental microarchitecture with unrivalled clarity [65].

Together, these modalities form a graduated diagnostic toolkit that allows clinicians to match imaging depth to case complexity and clinical resources.

Magnetic resonance imaging (MRI) is also emerging as an important diagnostic tool, especially for observing changes in soft tissues caused by dental issues in rabbits [14]. MRI offers exceptional clarity in depicting odontogenic abscesses and their connections to neighboring anatomical structures, especially when dealing with complex complications like retromasseteric and retrobulbar abscesses [59].

Rigid oral endoscopy (stomatoscopy) permits the in vivo inspection of occlusal surfaces, gingivae, and pulp exposure and provides guided access for biopsy or burring in otherwise hard-to-reach molar rows. In particular, for rabbits and rodents with small oral openings, rigid endoscopy offers precise illumination and magnification, which provide an exceptional view of the oral cavity [68]. Stomatoscopy, specifically for oral examination, is valuable in diagnosing dental diseases in rabbits due to its noninvasiveness, ability to inspect the oral cavity thoroughly, magnified visualization of dental surfaces, reduced risk of missing lesions, and facilitation of therapeutic procedures [69]. Because stomatoscopy visualizes the clinical crowns and gingiva, while cross-sectional imaging depicts the apical (reserve) crowns and surrounding bone, an accurate workup of rabbit dental disease must integrate careful physical/oral examination, rigid oral endoscopy, and advanced imaging (CT or CBCT). Only this combined approach reveals the full extent of pathology and guides appropriate treatment [65].

Microbiological culture is most informative when the sampling strategy matches the lesion biology. Because rabbit pus is often sterile, clinicians should submit both purulent material and a strip of the abscess capsule: viable bacteria typically reside at the peripheral lining rather than in the core [37,70]. Even so, the center yields additional organisms in roughly half of cases, so culturing both sites maximizes recovery [13]. Taking samples directly from the abscess or affected tissue, and promptly placing them in transport medium, minimizes contamination by normal gingival flora and therefore improves the clinical relevance of the culture and sensitivity results [51,59].

## 8. Complications and Prognosis

Retrobulbar abscesses and skull empyemas in rabbits are serious complications frequently associated with ADD, particularly involving the maxillary cheek teeth. The reserve crowns of maxillary cheek teeth 3 to 6 are situated inside a distinctive anatomical structure known as the alveolar bulla, which is located cranially, ventrally, and medially to the orbital fossa. This proximity facilitates the extension of periapical infections from these teeth into the retrobulbar or parabulbar spaces [59].

Retrobulbar disease is the most common cause of exophthalmia in rabbits and is typically unilateral. Retrobulbar abscesses commonly originate from odontogenic infections, although they may also result from penetrating foreign bodies, local trauma, or hematogenous spread [71,72,73]. These abscesses cause inflammation and increased pressure within the retrobulbar space, affecting the ocular muscles, optic nerve, and surrounding orbital tissues, increasing the risk of ocular trauma due to exophthalmos and subsequent vision loss. This pressure can lead to exophthalmia and potential ocular damage. In severe cases, the inflammation and pressure may necessitate enucleation. The management of secondary ocular disease—including enucleation when required—must be coupled with definitive treatment of the underlying retrobulbar dental abscess; otherwise, recurrence is likely [12]. Additionally, infection can extend laterally to involve the accessory lacrimal gland, resulting in a parabulbar abscess [59].

Empyemas of the alveolar bulla are a precursor to retrobulbar abscessation. In rabbits with advanced acquired dental disease (ADD), anatomical distortion of the maxillary cheek-tooth apices can create a void within the alveolar bulla. This space, normally occupied by spongy bone rather than air, may fill with food debris or become colonized by bacteria, forming an empyema (a purulent collection inside a pre-existing cavity) [59,74]. Once infected material breaches the thin bony wall of the bulla, it readily extends into the retrobulbar fat, giving rise to the clinically recognizable retrobulbar abscess. Thus, the treatment of ocular or orbital complications should always include investigation for and management of any underlying alveolar-bulla empyema [59].

Several practical factors should be taken into consideration when evaluating the prognosis: the quality of post-operative care and the owner’s understanding and compliance, as well as the cost of treatment. Due to enhanced diagnostic imaging, surgical techniques, and overall treatment strategies, the outlook has become more promising, even for complex cases, compared to a few years ago. Remarkable improvements can be seen in rabbits after facial surgery, and even chronic cases can be managed long-term while maintaining a good quality of life [69].

The prognosis should therefore be tailored to each patient and clinical case [59]. Accurate diagnosis is essential for evaluating the prognosis and planning an effective management strategy [62].

The prognosis for dental disease in rabbits varies depending on the primary cause and the extent of secondary changes. Congenital malocclusions often carry a guarded to poor prognosis due to their progressive nature and early onset. In contrast, traumatic causes may have a more favorable outcome if addressed promptly. Cases related to metabolic bone disease (MBD) often have a poorer prognosis due to diffuse skeletal involvement and compromised bone quality [12,75]. It is recommended to conduct bacterial cultures in rabbits with odontogenic abscesses in order to offer clients additional prognostic information [53].

The anatomical location also plays a key role in prognosis. When abscesses are located on the rostral mandible, in the lateral cheek, in the area in front of the medial canthus of the eye, or in the zygomatic region below the eye, the outlook is generally favorable. This is because either the apex of the teeth involved is accessible or the infection is localized. However, if the abscesses are located in the caudal mandible or involve the nasal cavity or paranasal sinuses, the prognosis becomes guarded due to the complexity of adjacent anatomical structures and the potential for extensive local invasion and spread [12].

While radiographic or CT evidence of osteomyelitis has traditionally been associated with a guarded to poor prognosis, recent findings suggest that successful outcomes are still possible with appropriate surgical and medical management [53]. In severe cases, such as those involving multiquadrant disease, a lack of response to repeated surgical and medical interventions, or persistent recurrence despite appropriate therapy, humane euthanasia should be considered. These cases are often associated with chronic pain, weight loss, and progressive osteomyelitis, even when the rabbit may appear alert or otherwise clinically stable [8]. Importantly, the decision to euthanize is always guided by the animal’s quality of life, in accordance with ethical standards and, in some countries, legal regulations. Quality of life does not always correlate directly with imaging findings or other diagnostic results, and clinical judgment must weigh all aspects of the patient’s welfare.

## 9. Treatment

### 9.1. Surgical Approach

Surgery is the cornerstone of treating rabbit odontogenic abscesses because antibiotics alone cannot penetrate the fibrous capsule, necrotic bone, or dense caseous pus that characterize these infections [59,76,77,78]. The ideal operation achieves three goals: (i) the en bloc excision of the capsule, (ii) the extraction of the diseased cheek tooth (including any reserve-crown fragments), and (iii) the aggressive curettage of osteomyelitic bone [59,76,79]. When anatomic constraints preclude complete removal, the surgeon should perform maximal debridement and then create a permanent drainage route—either marsupialization or wound-packing—to allow repeated flushing or topical therapy [59,62,80,81]. Pre-operative radiography or CT defines the lesion’s extent and guides the choice between full excision, marsupialization, or packing, especially when mandibular or maxillary bone is involved [37,70,79]. Tooth extraction is mandatory whenever the abscess originates from an infected apex [59,70]. In every case, the capsule and pus are submitted for culture; systemic antibiotics are started empirically and adjusted once the results return, while local adjuncts (e.g., PMMA beads or Manuka honey) support bacterial clearance [29,62,76,77,79]. This combined surgical-and-medical strategy offers the best chance of long-term resolution. Beyond surgical debridement and targeted antimicrobials, robust pain control (e.g., meloxicam 0.2–0.3 mg kg^−1^ PO q24 h or buprenorphine 0.03 mg kg^−1^ SC q8 h) and proactive nutritional support with high-fiber critical-care formulas are essential adjuncts; they minimize ileus, maintain body condition, and enhance overall recovery [36,46].

Overall, a combined approach that integrates one of the above surgical techniques with appropriate antimicrobial therapy offers the highest chance of clinical success and the long-term resolution of abscesses in rabbits [76,77,79].

### 9.2. Systemic Antimicrobial Therapy

Rabbit odontogenic abscesses almost always require antibiotics, yet choices are restricted by two factors: fatal enterotoxemia if gut flora are disrupted and the global drive to slow antimicrobial resistance (AMR). Oral β-lactams, macrolides, lincosamides, and many cephalosporins can precipitate dysbiosis and rapid death [77,82]; therefore, they are reserved for local delivery in PMMA beads or avoided altogether [12,78]. The few systemically tolerated classes—fluoroquinolones, potentiated sulfonamides, and, with caution, penicillin–β-lactamase-inhibitor combinations—should be chosen on the basis of aerobic and anaerobic culture and sensitivity whenever possible; thick, caseous pus sharply limits empirical success [29,59]. Fluoroquinolones (e.g., enrofloxacin) remain a first-line treatment because of predictable oral absorption and efficacy against *Pasteurella*, *Pseudomonas*, and *Mycoplasma* spp. [79], but they are also listed by the WHO as “Highest-Priority Critically Important Antimicrobials,” highlighting the need for stewardship [77,83].

Evidence is limited, but two small crossover studies showed that this strain can stabilize cecal micro-flora during enrofloxacin or trimethoprim–sulfa therapy and reduce the incidence of soft-stool episodes in pet rabbits [59,84]. Although probiotics do not influence the incidence of dental abscesses, administering a well-studied strain such as Enterococcus faecium NCIMB 30183 may help stabilize the hind-gut microbiota when prolonged, broad-spectrum antibiotics are required, thereby reducing the risk of antibiotic-associated enterotoxemia in susceptible rabbits [84]. Because broad-spectrum antibiotics predispose rabbits to dysbiosis and enterotoxemia, we advise offering a licensed probiotic preparation whenever systemic antimicrobials are prescribed while stressing to owners that data remain preliminary.

AMR is a top-ten global health threat that links the human, animal, and environmental spheres (One-Health concept) [85,86]; the cross-species transfer of resistant bacteria has already been documented. Given the narrow therapeutic margin in rabbits and limited drug arsenal, each empirical course should be justified by urgency, informed by local susceptibility patterns, and replaced or refined once laboratory results return [1,59,87].

Figure 2 is intended as an overview of emerging resistance patterns, not as a tool for selecting an empirical drug. Clinicians should still begin with first-line antibiotics that cover the core commensals/opportunists commonly cultured from rabbit abscesses and then adjust therapy once culture and sensitivity results identify less common or resistant organisms.

The heatmap reveals a concerning pattern of multidrug resistance in organisms such as *Stenotrophomonas maltophilia* and *Burkholderia* spp., which exhibited resistance rates of 85–90% across all tested antibiotics, including amikacin, gentamicin, enrofloxacin, penicillin, chloramphenicol, and trimethoprim–sulfamethoxazole [1]. These pathogens pose significant therapeutic challenges and often necessitate advanced culture-guided or localized treatment strategies.

On the other end of the spectrum, low or no resistance was reported for *Pasteurella multocida*, *Streptococcus* spp., and *Staphylococcus* spp. These organisms are frequently implicated in rabbit dental infections but remain largely susceptible to first-line antimicrobials, reinforcing their treatability and continued empirical coverage in early-stage abscess cases [1,51,52,53].

Intermediate resistance patterns were observed in *Enterobacteriaceae*, such as *Escherichia coli* (approx. 48%) and *Klebsiella pneumoniae* (approx. 58%), suggesting potential for partial susceptibility depending on the drug class and clinical context [1,51,52]. Notably, *Enterobacter cloacae* displayed consistent but moderate resistance (36%), highlighting the need for caution in empirical use [51].

Ultimately, this heatmap indicates the need for routine susceptibility testing, judicious antibiotic use, and ongoing surveillance to mitigate the rise of resistance and guide rational antimicrobial therapy in exotic pet medicine.

A significant complication in the management of odontogenic abscesses in rabbits is the presence of bacterial biofilms. These structured microbial communities adhere to necrotic bone, tooth fragments, and implanted materials, encased in a self-produced extracellular matrix that significantly impairs antibiotic penetration. As a result, biofilms contribute to recurrent or refractory infections, even when antimicrobial therapy is appropriately selected based on culture and sensitivity testing [53,88]. The chronic nature of these infections often necessitates repeated surgical debridement, the complete excision of infected tissue, and, in some cases, the use of local antibiotic delivery systems (e.g., AIPMMA beads) to achieve effective concentrations at the site of infection [32]. Additionally, adjunctive therapies aimed at disrupting biofilm architecture—such as enzymatic agents, antiseptic flushing, or honey with proven antibiofilm properties—may be beneficial [53,88,89]. Although anaerobes such as *Fusobacterium* spp., *Bacteroides* spp., and *Peptostreptococcus* spp. are known contributors to polymicrobial abscesses [13,51,53,54], their presence is likely underreported due to the technical challenges associated with anaerobic culture. Their omission in resistance data should not be interpreted as clinical insignificance but rather as a limitation in the diagnostic approach.

### 9.3. Local Antimicrobial Therapies in Rabbit Odontogenic Abscesses

With the increasing prevalence of multidrug-resistant (MDR) bacterial infections in companion animals, especially exotic pets such as rabbits, clinicians are exploring alternative and adjunctive therapeutic strategies. Among these, topical antimicrobial treatments—such as the use of natural products like honey or the application of antibiotic-impregnated polymethy lmethacrylate (AIPMMA) beads—are gaining attention for their ability to provide high local antimicrobial concentrations while minimizing systemic side effects [90].

#### 9.3.1. Honey as a Topical Antibacterial Agent

Given the challenges posed by odontogenic abscesses in rabbits, particularly those involving bone and soft tissue, adjunctive treatments that support wound healing and infection control are of notable interest. Honey has emerged as a potential topical agent in such cases due to its various biological properties [59,91].

Because of its high sugar content, honey has a hygroscopic effect on wounds, which inhibits bacterial growth. Additionally, it acidifies the wound, thereby accelerating the healing process. Honey can be applied directly to the wound surface or used as part of wound-packing materials, with reapplication typically recommended once daily, depending on exudate levels and clinical progression. Notably, honey offers the benefits of being non-toxic and having no reported side effects after ingestion through grooming in rabbits, which allows for its prolonged use in treatment protocols [56].

Of particular relevance in the context of odontogenic abscesses involving bone structures is honey’s potential role in bone regeneration. Its anti-inflammatory, antimicrobial, and antioxidant properties have been shown to speed up the inflammatory process and stimulate osteogenesis, thereby supporting recovery in cases complicated by osteomyelitis. Among various types, Manuka honey has demonstrated superior antimicrobial and antioxidant effects and has shown efficacy in vitro against pathogens commonly involved in abscess formation, such as *Staphylococcus aureus* and *Pseudomonas aeruginosa* [92,93].

The healing properties of honey are supported by its ability to stimulate macrophages, ensure rapid infection clearance, maintain wound sterility, and promote healthy tissue regeneration [94]. It promotes a moist wound environment, enhances the release of cytokines such as interleukin-1, interleukin-6, and tumor necrosis factor, and modulates inflammation by lowering prostaglandin levels and increasing nitric oxide. Moreover, honey’s protease enzymes facilitate autolytic debridement, support granulation and epithelization, and help reduce scar tissue formation, making it a valuable tool in managing complex abscesses [95,96].

#### 9.3.2. Antibiotic-Impregnated Polymethyl Methacrylate (AIPMMA) Beads

AIPMMA beads are created intra-operatively from antibiotic powder and PMMA cement, then implanted directly into the abscess cavity. They maintain high local drug levels for 2–4 weeks while avoiding the systemic side effects that frequently limit medical therapy in rabbits [13,29,59,97]. While commercially prepared AIPMMA beads are available in certain regions, particularly in the United States, they can also be manually produced in veterinary settings [83]. The preparation of AIPMMA beads poses several challenges. They are often handmade during surgery, leading to variability in bead size, shape, and antibiotic release rates. Larger beads may be more difficult to implant and can extend the surgical time. Notable disadvantages include their rigidity and the incapacity to modify treatment after implantation in response to culture results [44].

The three agents most widely used in practice are as follows:Gentamicin—This agent is heat-stable, commercially available in veterinary PMMA kits, and active against the *Staphylococcus*–*Pseudomonas* spectrum common to mandibular abscesses [98].Amikacin—This agent has broader Gram-negative coverage (including multidrug-resistant *Pseudomonas*) and is safe locally, even in animals with marginal renal function; the beads must lie flush against the bone because diffusion rarely exceeds 5 mm [32].Clindamycin—This agent has potent anaerobic activity and is also heat-stable but should be reserved for cases where the bead track is fully sealed, as systemic absorption can precipitate enterotoxemia in rabbits [56,98].

Beads must not communicate with the oral cavity or skin, or they may be chewed out or leaked and ingested. Once elution is exhausted, non-biodegradable PMMA can act as a nidus for reinfection; therefore, removal should be scheduled or at least monitored radiographically [59,97,99]. Heat-labile drugs such as penicillins or enrofloxacin lose activity during polymerization, and metronidazole remains experimental in microencapsulated form [100]. When culture results dictate an alternative antibiotic, new beads can be mixed and exchanged during a second-look procedure. 

Local delivery also permits the use of agents, such as gentamicin, amikacin, lincomycin, and ceftazidime, which are poorly tolerated or even contraindicated when given systemically in rabbits because they disrupt the hind-gut flora or potentially carry a nephrotoxic risk [90].

Table 1 provides a comprehensive overview of antibiotics assessed for their compatibility with antibiotic-impregnated polymethyl methacrylate (AIPMMA) beads, an increasingly used method for localized antibiotic delivery in the treatment of rabbit odontogenic abscesses. The compatibility of these antibiotics (bolded) is determined primarily by their heat stability, as the polymerization of PMMA is an exothermic process that can degrade heat-labile agents. This characteristic is critical to ensure that the embedded antibiotic retains its antimicrobial efficacy after being incorporated into the bead matrix.

Heat-stable antibiotics such as gentamicin, amikacin, tobramycin, and ceftazidime maintain their activity after polymerization and are widely used in both human and veterinary medicine, particularly for deep-seated infections. Their broad-spectrum efficacy, especially against *Pseudomonas aeruginosa*, makes them ideal for local delivery via AIPMMA beads, achieving high concentrations with reduced systemic risks in rabbits [12,32,78]. Clindamycin is conditionally compatible; although heat-stable and effective against anaerobes, its systemic absorption can trigger enterotoxemia in rabbits, so its use should be restricted to cases where local release is unlikely to cause systemic exposure [56,98]. In contrast, heat-labile antibiotics like enrofloxacin, penicillins, and standard metronidazole are unsuitable for AIPMMA use due to inactivation during polymerization. Research into microencapsulation may expand future options, but such technologies are not yet clinically available [32,100]. This compatibility matrix serves as a practical guide for clinicians, ensuring that antibiotic selection is based not only on microbial sensitivity patterns but also on the physicochemical properties of the delivery system. Given the increasing prevalence of multidrug-resistant pathogens in rabbit odontogenic infections, the ability to deliver effective antibiotics directly to the infection site while minimizing systemic toxicity is a valuable strategy. The use of AIPMMA beads, when guided by compatibility considerations, as outlined in Table 1, allows for precision-targeted therapy, reducing recurrence and improving treatment outcomes in complex or refractory cases.

Figure 3 illustrates an essential clinical decision-making algorithm for managing rabbit odontogenic abscesses using local antimicrobial therapies. The flowchart begins with essential diagnostic steps, including the confirmation of bone and/or soft-tissue involvement and culture and sensitivity testing. Because osteomyelitis and chronic, caseous infection are almost universal features of rabbit odontogenic abscesses, local antibiotic delivery (e.g., AIPMMA beads) is generally indicated; these factors reinforce, rather than determine, the decision to use a local antimicrobial approach.

A critical decision point in the algorithm assesses whether systemic antibiotic therapy is contraindicated, which can occur due to risks of gastrointestinal dysbiosis, enterotoxemia, or limited systemic efficacy in necrotic tissue [12,78]. If systemic use is not advised, the flowchart guides the clinician toward localized treatment options such as topical honey or AIPMMA beads, selected based on the nature and extent of tissue involvement [29,91,97].

Topical honey, particularly medical-grade Manuka honey, is recommended when primarily soft tissue is affected. It acts as an adjunct by promoting granulation tissue, wound healing, and providing antimicrobial effects. This option is especially useful for non-toxic, grooming-safe, and long-term applications [92,93,94].

AIPMMA beads are indicated when there is osteomyelitis or a need for sustained high local antibiotic delivery, especially in cases involving bone. These beads are implanted intra-operatively near the abscess or necrotic site after debridement and provide prolonged local antimicrobial exposure. The selection of heat-stable antibiotics (as listed in Table 1) is essential to ensure efficacy post polymerization. Monitoring is necessary to assess for foreign body reactions, and bead removal may be considered after drug elution [32,90,98].

If systemic therapy is not contraindicated, AIPMMA beads can still be used as an adjunct to systemic antibiotics, particularly in complex or refractory cases [76,77].

Post-treatment complications can occur. After surgical curettage or tooth extraction, rabbits may develop anemia from intra-operative hemorrhage, orocutaneous fistulae that complicate post-operative feeding, iatrogenic jaw fractures, and secondary malocclusion; recurrence of the abscess remains the most frequent late complication [12].

The flowchart concludes with post-treatment considerations, emphasizing incision closure to prevent grooming-related contamination, the prophylactic use of probiotics (especially with clindamycin-like antibiotics), and the need for ongoing monitoring of wound healing and systemic recurrence [56,84]. This clinical algorithm supports individualized treatment planning, optimizing outcomes by guiding antimicrobial strategies according to the severity, location, and microbiological characteristics of the abscess.

## 10. Practical Considerations for General Veterinary Practice

### 10.1. Diagnostic Prioritization Without Advanced Imaging

In settings where CT or CBCT is unavailable, clinicians should still pursue a structured diagnostic approach. Thorough extraoral palpation and intraoral examination under sedation can provide valuable clinical insight, particularly when complemented by a full series of high-quality skull radiographs. A minimum of four to five standard projections, including lateral, right and left lateral oblique, dorsoventral, and, occasionally, rostrocaudal views, are recommended to overcome anatomical superimposition and allow the accurate assessment of dental and bony pathology [13,35,63,69].

While plain radiographs have limited sensitivity, they can still reveal important indirect indicators such as tooth root elongation, periapical radiolucency, mandibular bone lysis or thickening, and changes suggestive of nasolacrimal duct displacement. However, the direct visualization of the nasolacrimal duct and confirmation of displacement require a contrast dacryocystogram [27,35,62].

In cases with clinical suspicion but equivocal radiographs, empirical therapy should be approached with caution, and referral for advanced imaging should be considered if recurrence occurs or surgical margins are unclear [1,29].

### 10.2. Optimizing Microbial Culture: Focus on Anaerobes

Anaerobes are underreported in rabbit odontogenic abscesses, largely because they are missed at the sampling stage [29,51]. When surgery is scheduled, the gold-standard specimens are a strip of abscess capsule plus any purulent material obtained intra-operatively. Percutaneous needle aspiration is reserved for situations in which immediate surgery is impossible or when a preliminary culture is needed to guide empirical therapy before definitive debridement. For either method, the anaerobic yield can be maximized by (i) sampling peripheral tissue rather than the superficial surface, (ii) using anaerobic transport media (e.g., Port-A-Cul^®^, Cary-Blair), (iii) capping syringes without delay, and (iv) explicitly requesting both aerobic and anaerobic cultures at the laboratory [13,37,70].

If culture is not feasible, choosing broad-spectrum antibiotics effective against anaerobes (e.g., metronidazole, clindamycin—used cautiously in rabbits) may be warranted for refractory or polymicrobial cases [8,37,51].

## 11. Prophylaxis in Rabbit Dental Health

Preventive strategies are essential in reducing the incidence and severity of odontogenic abscesses in pet rabbits. Given their elodont, aradicular teeth that continuously grow, a preventive approach must center on environmental, nutritional, and behavioral factors. This approach should include the following:Dietary management: The cornerstone of prophylaxis is dietary fiber. Feeding a hay-based diet (timothy, meadow, or orchard grass) promotes prolonged mastication, which supports dental wear and overall oral health. Pellets should be offered in limited amounts, and muesli-type commercial mixes should be avoided due to their association with selective feeding and malocclusion. A calcium-balanced diet remains important—not because dietary hypocalcemia is a common issue in rabbits but due to the ongoing demand for calcium in continuously growing teeth and bone. However, this does not imply a need for high calcium or vitamin D intake, as excessive supplementation may lead to serious health problems such as hypervitaminosis D and renal pathology. Notably, rabbits can maintain normal serum calcium concentrations even in the presence of metabolic bone disease through compensatory mechanisms such as increased parathyroid hormone activity, which mobilizes calcium from the skeleton. This bone resorption, in turn, contributes to alveolar bone loss and dental instability [41,46,62,101].Regular comprehensive examinations: Every 6–12 months, rabbits should receive a full physical exam plus a conscious intraoral inspection with an otoscope or small speculum. In high-risk cases (e.g., lops, dwarfs, or rabbits with prior dental disease), stomatoscopy, a rigid endoscopic examination of the oral cavity performed under general anesthesia, provides a magnified view of cheek-tooth arcades and should be scheduled at longer intervals (e.g., every 12–18 months) or sooner if clinical signs appear [59].Environmental enrichment: Providing fibrous forage and appropriate items for oral activity supports a normal masticatory pattern, which is essential for even tooth wear, particularly of the cheek teeth. While chewable items such as untreated wood blocks or willow branches primarily engage the incisors, long-stemmed hay and leafy greens are critical for promoting proper lateral jaw movement and physiological attrition of the cheek teeth [46,70,102].Owner education: Prophylactic success hinges on owner compliance. Educating clients on the importance of nutrition, the signs of early dental disease (e.g., epiphora, weight loss), and the dangers of improper trimming (e.g., with nail clippers) is crucial [103].Preventing iatrogenic injury: Dental procedures must be performed by trained professionals. Improper burring or incisor trimming can result in pulp exposure, leading to infection and abscess formation [59].

A multi-pronged prophylactic strategy combining dietary control, regular monitoring, and client education can significantly reduce the incidence of odontogenic abscesses and enhance rabbit welfare.

## 12. Discussion

Rabbit odontogenic abscesses are among the most challenging clinical conditions encountered in exotic animal practice. Their complex microbiology, insidious progression, and anatomical location demand a multifaceted diagnostic and therapeutic approach. This review compiles recent evidence on microbiological profiles, imaging, treatment modalities, and the emerging role of local antimicrobial therapies.

The integration of comparative heatmaps in this review offers novel insights into the complex microbiological landscape of rabbit odontogenic abscesses. The microbial profile of rabbit odontogenic abscesses is characteristically polymicrobial, involving both aerobic and anaerobic species [29,51]. The first heatmap builds upon this microbiological perspective by mapping the reported presence of bacterial species across six major studies, including a recent Romanian study by Crăciun et al. (2025) [52]. This comparative visualization accentuates the consistency of core pathogens (*Staphylococcus* spp., *Pasteurella multocida*, *E. coli*) across geographic regions while also introducing emerging organisms such as *Glutamicibacter protophormiae*. The inclusion of this rare isolate—first documented in Romanian pet rabbits—suggests that the bacterial diversity in odontogenic abscesses may be broader than previously recognized and may be highly influenced by regional and environmental factors. Additionally, the inconsistent detection of anaerobes such as *Fusobacterium* spp. and *Peptostreptococcus* spp. may reflect limitations in diagnostic protocols rather than true absence.

The second heatmap, presenting antibiotic resistance profiles across key bacterial species, reveals a growing prevalence of multidrug-resistant (MDR) organisms. Notably, pathogens such as *Pseudomonas aeruginosa*, *Enterobacter cloacae*, and *Stenotrophomonas maltophilia* exhibit high resistance rates to commonly used antimicrobials, highlighting the limitations of empirical antibiotic therapy in the absence of culture and sensitivity data [1,51]. Conversely, organisms like *Pasteurella multocida* and *Streptococcus* spp. continue to show relatively low resistance profiles, reinforcing their role as primary but manageable culprits [1,51,52,53].

Accurate diagnosis is essential in managing rabbit odontogenic abscesses. Traditional radiographs often miss early lesions, while CT and CBCT offer superior detail [65]. CBCT is especially useful due to its low radiation, high bone contrast, and accessibility, allowing the clear visualization of abscess extent, bone changes, and adjacent structures, as well as the monitoring of post-treatment outcomes [45].

When considered alongside the treatment flowchart, these microbial findings confer support to a more nuanced, multimodal treatment strategy. Given the limited vascularization of abscess capsules and the dense, caseous consistency of rabbit pus, systemic antibiotics alone are often inadequate [57,82]. Figure 3 summarizes how local antimicrobial adjuncts—such as Manuka-honey dressings or antibiotic-impregnated PMMA beads—should be selected after the surgical approach (excision, marsupialization, or wound-packing) has been chosen and culture and sensitivity samples have been obtained. These strategies aim to bypass systemic limitations, achieving high localized antibiotic concentrations while minimizing systemic toxicity.

Various treatment approaches have been reported for odontogenic abscesses in rabbits, with different outcomes. A retrospective study on 13 rabbits showed that minimal debridement combined with weekly antimicrobial gauze-packing and systemic antibiotics resulted in the resolution of 13 out of 14 abscesses over a 32.6-month follow-up period [29]. In another study, seven out of nine rabbits with retrobulbar abscesses treated surgically and with long-term antibiotics showed no recurrence for at least six months [53]. A larger study involving 200 rabbits reported a recurrence rate of just 8% following extraoral tooth extraction, marsupialization, and radical bone debridement guided by CT, with a median disease-free interval of 29 months [54]. However, rabbits treated with antibiotics alone had a lower long-term success rate, highlighting the importance of surgical intervention for sustained outcomes [54].

Topical honey, particularly medical-grade Manuka honey, is recommended when primarily soft tissue is affected. It acts as an adjunct by promoting granulation tissue, supporting wound healing, and providing antimicrobial effects [92,93,94]. Notably, Manuka honey also exhibits osteogenic properties, making it valuable in cases involving bone loss or osteomyelitis [59,91]. Its non-toxic nature and safety when ingested through grooming allow for prolonged application, especially in rabbits.

Similarly, antibiotic-impregnated polymethyl methacrylate (AIPMMA) beads are indicated when osteomyelitis is present or when high local antibiotic delivery is required. These beads are implanted intra-operatively near the abscess or necrotic site following debridement, ensuring sustained, site-specific antimicrobial release [32,90,98]. They are particularly useful in refractory infections, including those caused by anaerobes or resistant Gram-negative organisms [13,29,59,97]. Heat-stable antibiotics (as listed in Table 1) must be selected to preserve efficacy post polymerization.

Monitoring is necessary to assess for potential foreign body reactions, and bead removal may be considered once drug elution is complete. When systemic therapy is not contraindicated, AIPMMA beads can be used adjunctively to enhance outcomes in complex or recurrent cases [76,77]. Post-treatment management should include incision closure to prevent grooming-related contamination, the prophylactic use of probiotics, especially when clindamycin-like antibiotics are employed, and regular monitoring to evaluate wound healing and detect any signs of systemic recurrence [56,84].

Non-surgical or minimally invasive management can be used for inaccessible abscesses. When the capsule cannot be reached safely, most commonly in retrobulbar, alveolar-bulla, or deep nasal locations, the goal shifts from complete excision to long-term infection control. Options include (i) creating a small osseous window to allow periodic flushing and bead replacement, (ii) the image-guided injection of antibiotic gel or Manuka-honey paste, and (iii) prolonged systemic therapy based on serial culture results. In retrobulbar disease, enucleation combined with curettage of the periorbital abscess wall can reduce mass effect and pain even if microscopic foci remain. The prognosis is guarded: recurrence rates exceed 60%, and owners should be counselled that repeated interventions or, in some cases, palliative management may be necessary [12,74].

Nonsteroidal anti-inflammatory drugs are the primary option for post-surgical pain control in rabbits, but gabapentin may be a useful adjunct in cases of chronic or neuropathic pain. Recent evidence supports its potential to improve comfort in rabbits with persistent or refractory discomfort [104].

Although rabbit odontogenic abscesses are typically non-transmissible between species, the microbial spectrum associated with these infections includes several opportunistic and zoonotic pathogens, such as *Pasteurella multocida*, *Staphylococcus aureus*, *Escherichia coli*, and occasionally *Pseudomonas aeruginosa*. Immunocompromised individuals, children, and those with direct exposure to infected purulent material (e.g., during at-home wound care) may be at increased risk of localized or systemic infection [105,106,107].

Moreover, the frequent detection of multidrug-resistant (MDR) strains in companion animals, including rabbits, raises broader One-Health concerns. As shown in the resistance heatmap, high resistance rates to aminoglycosides and fluoroquinolones, which are also critical in human medicine, highlight the importance of judicious antimicrobial use, guided by culture and sensitivity testing. These findings support ongoing antimicrobial stewardship efforts across veterinary and human medicine, reinforcing the need for the coordinated monitoring of resistance patterns and prudent prescribing practices in exotic pets [108,109,110,111].

Several of the organisms isolated from rabbit odontogenic abscesses, most prominently *Pasteurella multocida*, *Staphylococcus aureus*, and opportunistic *Enterobacterales*, also feature in human wound and respiratory infections, highlighting a potential, though low-incidence, zoonotic risk, especially for immunocompromised owners. The identification of many isolates resistant to β-lactams and fluoroquinolones, a pattern also seen in human dental cases and other pets, shows that antibiotic resistance is a shared One-Health issue. Continuous culture-based surveillance in companion mammals, harmonized breakpoints, and judicious, culture-guided antibiotic use are essential to curb the cross-species dissemination of resistant clones. Consequently, empirical therapy should avoid critically important antimicrobials unless susceptibility data or clinical failure necessitate their use [46,108,112,113,114].

While this review emphasizes the importance of adequate calcium and vitamin D levels in maintaining dental and skeletal health, caution is warranted to avoid misinterpretation. The excessive intake of calcium or vitamin D can lead to serious health problems in rabbits, including vitamin D toxicity and kidney damage. Therefore, the term “adequate intake” should not be understood as encouraging high supplementation, especially by readers less familiar with rabbit physiology. It is also important to note that vitamin D levels in rabbits are not solely determined by diet. Environmental factors, such as exposure to natural or artificial UVB light, play a significant role in vitamin D synthesis. Recent studies have demonstrated that UVB exposure—either directly to the animal or to its food—can influence the vitamin D status in rabbits [115,116,117]. These findings highlight the complexity of calcium and vitamin D metabolism in this species and underline the importance of a balanced, evidence-based approach to their nutritional management.

This comprehensive view of microbial patterns and therapeutic modalities reinforces the need for individualized care in rabbit dentistry. It also encourages the adoption of evidence-based protocols tailored to local resistance trends and species prevalence. Veterinary practitioners are thus encouraged to pursue culture and sensitivity testing whenever feasible, employ advanced imaging to guide surgical planning, and consider adjunctive therapies as part of a comprehensive treatment regimen.

Ultimately, this review highlights the importance of harmonizing diagnostic, microbiological, and surgical perspectives to manage rabbit odontogenic abscesses more effectively. The incorporation of comparative data visualizations not only enhances diagnostic precision but also guides clinicians toward rational, impactful therapeutic decisions that can improve both the prognosis and long-term quality of life for affected rabbits.

## 13. Gaps in the Literature and Research Needs

Despite increased awareness and improved diagnostics, substantial knowledge gaps remain in the study of odontogenic abscesses in rabbits:Limited data on anaerobic bacteria: While anaerobes such as *Fusobacterium* spp., *Peptostreptococcus* spp., and *Bacteroides* spp. are known contributors to abscesses in other species, their detection in rabbits remains rarely reported due to limitations in anaerobic culture and diagnostic protocols. Future studies employing molecular tools (e.g., 16S rRNA sequencing, metagenomics) are needed to clarify their role in polymicrobial infections.Lack of standardized treatment protocols: Most rabbit abscess treatments are empirical or based on isolated case reports. There is a clear need for randomized clinical trials comparing surgical, systemic, and local therapies, as well as long-term outcome data.Insufficient data on long-term outcomes: Few studies track rabbits beyond the immediate post-operative period. Long-term recurrence rates, quality of life, and complication rates (e.g., from AIPMMA bead retention) need evaluation.Underrepresentation of regional pathogens: Most microbiological studies originate from Western Europe and North America. There is a lack of data from Asia, Eastern Europe, and developing regions, despite evidence of regional variability in bacterial isolates.Diagnostic imaging thresholds: The criteria for when to escalate from radiography to CT or CBCT remain unclear. Studies are needed to define cost–benefit thresholds, especially for general practitioners working in resource-limited settings.Manuka honey and natural therapies: Despite promising in vitro data, there is a lack of controlled in vivo trials on honey’s effects on wound healing, osteogenesis, and recurrence prevention in rabbits. More research is needed to validate dosing protocols and establish safety over prolonged treatment periods.

## 14. Conclusions

Odontogenic abscesses in rabbits present a complex clinical challenge due to their unique dental anatomy, caseous abscess formation, and often chronic nature. Early detection through clinical evaluation and advanced imaging significantly improves prognostic outcomes. While surgical intervention remains the mainstay of treatment, adjunct therapies like topical Manuka honey and AIPMMA beads represent promising advances, particularly in refractory or osteomyelitic cases. Veterinarians must consider individual anatomical, etiological, and microbiological factors when designing treatment plans. Ongoing research into minimally invasive diagnostics and innovative localized treatments holds the potential to transform the clinical management of this common exotic pet condition.

## Figures and Tables

**Figure 1 animals-15-01994-f001:**
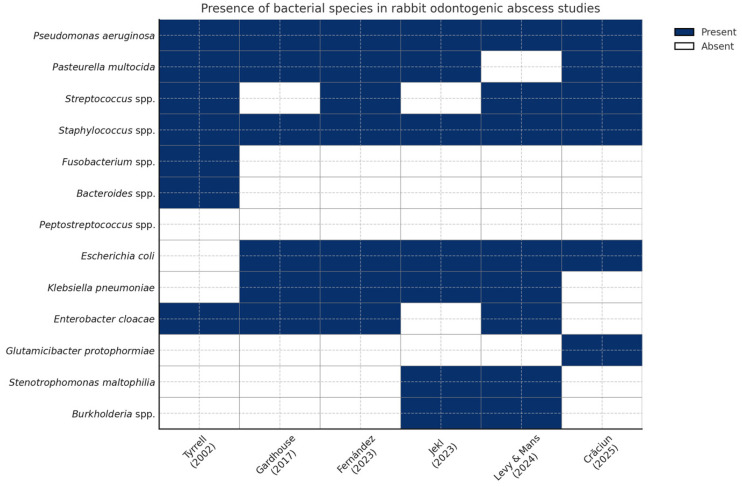
Heatmap showing the presence (blue) or absence (white) of bacterial species cultured from rabbit odontogenic abscesses in six peer-reviewed studies (columns ordered chronologically, 2002 → 2025) [1,13,51,52,53,54]. The grid makes it easy to spot both consistently reported pathogens and rarer or emerging isolates.

**Figure 2 animals-15-01994-f002:**
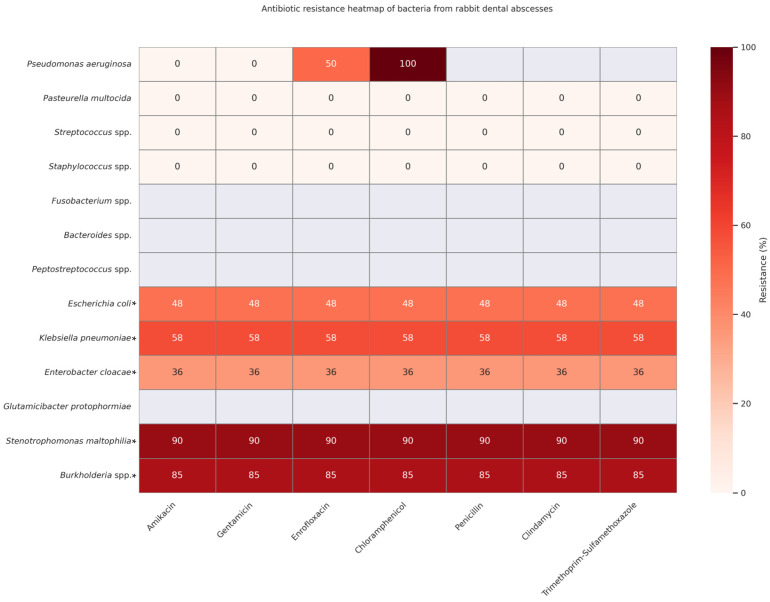
Antibiotic resistance heatmap of 421 bacterial isolates cultured from rabbit odontogenic abscesses in six peer-reviewed studies—Fernández et al., 2023 [1], Tyrrell et al., 2002 [13]; Gardhouse et al., 2017 [51]; Crăciun et al., 2025 [52]; Levy & Mans, 2024 [53], and Jekl et al., 2023 [54]. Cells are color-coded from white (0% resistant) to dark red (100% resistant). For each species–drug pair, % R = (number of resistant + intermediate isolates ÷ number tested) × 100; if a study did not test a drug, it was excluded from that denominator. Rows marked with an asterisk (*) and shaded light grey originate from a single dataset that used the same seven-drug panel for all *Enterobacterales* isolates; hence, these rows share identical percentages.

**Figure 3 animals-15-01994-f003:**
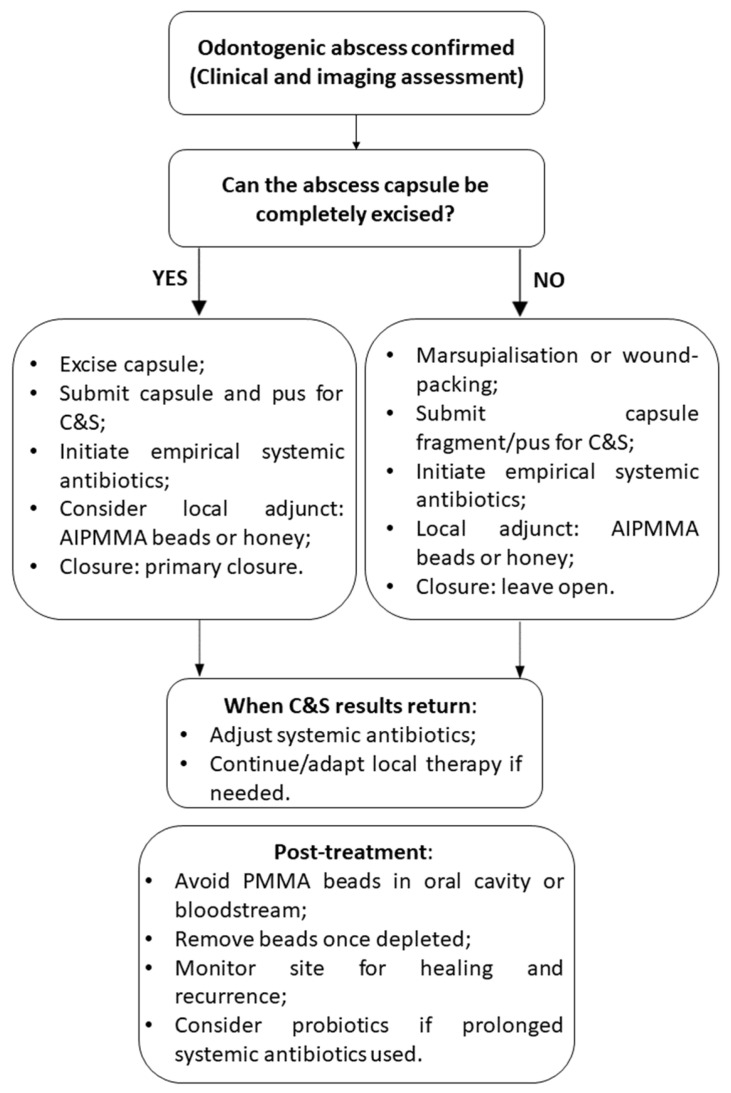
Treatment flowchart illustrating the local antimicrobial therapy approach for rabbit odontogenic abscesses.

**Table 1 animals-15-01994-t001:** Compatibility of antibiotics with AIPMMA and their clinical relevance.

Antibiotic	Compatible with AIPMMA?	Remarks
**Gentamicin**	Yes	Heat-stable. Released via concentration gradient, achieving higher local levels than systemic levels. Effective against Gram-negatives. Common in mandibular infections [32,98].
**Amikacin**	Yes	Heat-stable. Effective against Gram−, including *Pseudomonas* spp. Ideal for local use due to systemic nephrotoxicity [32,98].
**Tobramycin**	Yes	Heat-stable. Similar spectrum to gentamicin. Used in both human and veterinary medicine [32].
**Neomycin**	Yes	Heat-stable. Narrower spectrum but useful against Gram−. Toxic systemically and, hence, good for local use [98].
**Cefazolin**	Yes	Heat-stable. Active against Gram+; stable during polymerization [32].
**Ceftazidime**	Yes	Heat-stable. Broad-spectrum, including *Pseudomonas* spp. Suitable for co-infections [32].
**Ceftiofur**	Yes	Heat-stable. Commonly used in veterinary medicine. Active against Gram− bacteria [32].
**Lincomycin**	Yes	Heat-stable. Similar spectrum to clindamycin. Useful against anaerobes, with similar risks for gut flora [98].
**Clindamycin**	Yes, with caution	Heat-stable. Good local release; active against anaerobes. Risk of enterotoxemia in rabbits if ingested accidentally [56,98].
Enrofloxacin	No	Heat-labile. Degrades during polymerization. Not recommended [32].
Penicillins	No	Heat-labile (e.g., ampicillin, penicillin G). Inactivated during PMMA polymerization [32].
Metronidazole	No (partially)	Heat-labile. Microencapsulated versions under research; not compatible in standard form [100].

## Data Availability

No new data were created.

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
