# Peer review of "Odontogenic Abscesses in Pet Rabbits: A Comprehensive Review of Pathogenesis, Diagnosis, and Treatment Advances"

_animals, 2025, doi:10.3390/ani15131994_

Round 1
Reviewer 1 Report
Comments and Suggestions for Authors
General description:
The manuscript entitled „Odontogenic abscesses in pet rabbits: a comprehensive review of pathogenesis, diagnosis, and treatment advances” presents a clear and useful overview of dental abscesses in pet rabbits, in the areas of clinical, imaging, surgical, and microbiology aspects. The topic is current and important, especially with major attention to exotic pets, antibiotic resistance, and modern treatment options. The authors bring together existing research with new visual elements such as heatmaps and flowcharts, making the review informative, practical, and useful for everyday vets and specialists. The original figures based on peer-reviewed studies from recent years are a strong point and help make the paper easier to understand.
General comments:
The topic is significant and looks at a part of veterinary medicine that hasn’t been covered in detail. The paper is well structured, easy to follow, and brings together recent studies and up-to-date treatment options. It emphasizes the importance of CT and CBCT scans, the difficulties of growing anaerobic bacteria, and why local antibiotic treatments like AIPMMA beads are highly valued.
The color heatmaps, each based on results from six studies, help make the information clearer and more interesting. The review also points out where more research and attention are needed, like the lack of data on anaerobic bacteria and the need for standard treatment methods.
There are a few general problems to address, such as:
- Clarify in the methods or figure captions how data across the six studies was selected and standardized. Explaining inclusion criteria or how inconsistencies were handled (e.g., resistance cutoffs, antibiotic panels) will improve transparency.
- I suggest briefly mentioning that pain control and nutritional support are essential elements associated with the treatment. This is not the focus of the review, but may demonstrate holistic case management awareness.
- The updated figures are visually engaging, but consider increasing label contrast or adjusting font weight.
Specific comments:
- Table 1 (Antibiotic compatibility with AIPMMA): You may consider bolding the names of fully compatible antibiotics for visual clarity.
- Throughout manuscript: double-check that all genus and species names (e.g., Pasteurella multocida, Pseudomonas aeruginosa) are in italics, and that spp. is not italicized.
- Discussion, final section: a short but focused comment on zoonotic potential and One Health antimicrobial resistance adds valuable broader context.
- References: Make sure final formatting aligns precisely with journal style.
Recommendation: Minor revisions
This review brings new and useful information, especially for vets working with exotic animals. It’s practical, easy to understand, and nearly ready for publication. With just a few small fixes to the text and formatting, it will be a great resource for both veterinarians and researchers, especially those interested in antibiotic use and better imaging tools in rabbits and other small mammals.
Author Response
Dear Reviewer,
Thank you for your thoughtful and constructive review. Your insights have been invaluable in refining and strengthening our manuscript. Below, we provide a detailed, point-by-point response to each comment, outlining the changes that have been made.

Reviewer 2 Report
Comments and Suggestions for Authors
General comments:
Thank you for taking time to produce this very comprehensive review. Although the majority of the information included is appropriate and valid, there is a lot of repeated information and information that is poorly organized. This is part of the reason why this manuscript is overly long.
Please also be aware that consistency in language is needed. For example, “odontogenic abscess” (used 26 times) was used interchangeably with “dental abscess” (used 23 times). They refer to the same condition. Professional and up-to-date nomenclature should also be used, some of which has been highlighted in my comments. In particular, “tooth root” used throughout the manuscript does not really apply to rabbit dental anatomy. You have stated in beginning (L65), that rabbit teeth lack an anatomical root.
This following comment is not a personal criticism, but an observation based on how the information was presented within the manuscript. The current version of the manuscript reads as though the authors have limited experience in clinical rabbit medicine and surgery. If this was not the case, I apologize. However, if this was the case, I would strongly recommend the revised version of this manuscript is reviewed by an exotic animal specialist with extensive experience in rabbit medicine, particularly dentistry, before resubmission.
L24: “synthesizes” vs. “collates”
L26-27: Predisposing factors should be the same as those for rabbit odontogenic disease i.e. congenital conformation, inappropriate diet (insufficient abrasiveness, calcium, Vit D, etc.), trauma, neoplasia
L30: “…emerging use of local antimicrobial delivery”. I would not consider the use of AIPMMA beads in the treatment rabbit odontogenic abscess emerging. This has been described and used for this purpose (personally by this reviewer) for over 20 years.
L34: “dietary fiber intake” vs. “optimal diet” given there is more than just fiber which promotes dental health in rabbits.
L48-50: A statement regarding the frequency of observed odontogenic disease in pet rabbits is needed here to explain why it is important for the veterinary professionals to be well versed in the pathophysiology and treatment of this condition. This statistic has been investigated in a number of literature (e.g. O’Neill et al, 2020, Mäkitaipale et al, 2015)
L53-55: Are some of these conditions (e.g. problems with the nasolacrimal duct) strictly dental issues? Or should they be considered secondary conditions? The most common nasolacrimal duct disease associated with dental disease in rabbits is chronic dacryocystitis/dacryosolenitis – this may be a more appropriate terminology here. If secondary conditions are listed, what about paranasal sinusitis, exophthalmos, etc.? Be also aware of consistency in vocabulary used – is “overgrowth” the same as “excessive growth”?
L55: Please list the original reference for this statement (Schumacher, 2011). Also, note that this study encompassed only 40 rabbits in total. I think it is prudent not to make absolute statements such as this in a review article. Consider modifying this sentence to something similar to, “It is noteworthy that up to 60-65% of young healthy rabbits has been shown exhibit dental abnormalities in one study”. It is also important to note that in that study, a significant portion of these rabbits did not show progression to pathology, so some of these abnormalities were concluded to be a variation of normal.
L59-60: In addition to diagnosis of dental disease, what else would the understanding of the the rabbit’s dental anatomy and physiology be essential for? Consider adding treatment and prognosis here too. I would also consider the understanding of the biomechanics of mastication essential to treatment and prognosis.
L87: Please elaborate on what you mean by “abnormal oral behavior”?
L90-91: Please expand and describe which of the two categories of rabbit dental disease (i.e. congenital vs acquired) is considered more common in pet rabbits.
L104-146: This section can be misleading for the less informed reader. Out of all the potential etiologies of dental disease in rabbits, which of the etiology/etiologies are considered most common and the primary cause? Metabolic bone disease and dietary factors (leading to reduced attrition/wear of teeth) are generally considered to be the two major contributing factors to dental disease in rabbits. These should be listed and discussed first. Please also include the major reference Jekl & Redrobe (2013) here.
L104-107: How common is periodontitis in rabbits (i.e. inflammation of the periodontal space as described)? Does this occur as a primary condition or does it occur more commonly as part of pathophysiology of PSADD?
L112-113: How does TMJ subluxation/luxation, mandibular fractures lead to odontogenic abscess formation if there were no direct impact or involvement to odontogenic structures. Could this be due to malocclusion and changes to supporting alveolar bone > altered masticatory biomechanics > further alterations in attrition and bone structure > widening of periodontal spaces > periapical infection? Please include a statement regarding the likely pathophysiology to improve clarity.
L135: Could you please provide some common examples of a diet low in abrasive particles?
L139-140: The syntax in this sentence is a little strange. Consider rephrasing.
L141-143: Please elaborate on the potential role of MBD in the etiopathogenesis of dental disease in rabbits. Is it more likely to be nutritional (vs. renal)? Briefly discuss the unique metabolism of calcium in rabbits.
L152: Is apical elongation of the maxillary incisor the only odontogenic cause of epiphora? What about apical elongation of the first maxillary premolar/cheek tooth?
L114: “tooth root abscess” – please modify the terminology used here given you have just explained earlier that rabbit teeth lack anatomical roots.
L147-161: While this section is relevant, its inclusion here is out of place. It needs to be incorporated earlier as part of the discussion on the etiopathogenesis of odontogenic abscesses
L171-176: It isn’t clear why you have separated the reporting of bacterial isolates from odontogenic abscesses into two groups. Are the results from “previous reports (L175)” considered sub-par?
L178-180: I think this sentence should be stated first in this paragraph.
Figure 1: A heat map denotes the use of colors to represent values (in this case the presence vs absence) in a set of data. Given the options are yes (1) or no (0), I’m not certain of the benefits in adding a number into each cell. It may be better that a legend is provided on the side, explaining that blue = presence, and white = absence.
L185-186: The Fernandez (2023) study looked at bacterial infections in general in pet rabbits. While the study looked at the bacterial isolates from abscesses and dental disease, it did not specify how these two categories were separated. Which set of data was used in your heatmap? Also, regardless of the set of data used, Pseudomonas was isolated in both categories. Why was this considered absent in the heatmap (yet Klebsiella was considered present when it comprised a smaller proportion of bacterial isolates identified)?
L205-206: The statement would be improved if you are able to order the studies chronologically on the heatmap.
L221: How are you differentiating between ocular discharge and dacryocystitis based on physical examination (as you described these as “most frequently observed signs”)?
L228: “dead cells”- colloquial. Consider necrotic cells
L233: Cytological examination of what? I assume purulent exudate?
L246-247: Precise palpation vs. detailed palpation. Dental “roots” – please see earlier comments regarding use of this terminology. Are these the only structures related to odontogenic disease that can be evaluated by conscious examination/palpation? What about the facial tuberosity, the lateral portion of the maxilla (periapical abscesses pertaining to the first and second maxillary cheek teeth can frequently manifest in these areas), retropulsion of the globe (which may represent retrobulbar abscesses if this is reduced)?
L251: Just dental structures or the oral cavity as well?
L251-253: This is a repeat of a sentence earlier. Also, are these dental “issues”? Or are they presentations of dental disease?
L254-256: Use of radiocontrast in imaging studies may also be required.
L257-258: Briefly discuss whether rabbit odontogenic abscesses are typically associated with any hematological changes.
L258-260: Ultrasonography should be included when discussing diagnostic imaging in the earlier paragraph, not here.
L261: Plain radiography vs. radiographic imaging
L264: I disagree with this. I think knowledge of rabbit dental disease and its diagnosis has significantly evolved since this publication in 2009. The lateral view particularly when assessing the origin of periapical abscesses associated with mandibular or maxillary cheek teeth is often limited by superimposition. Lateral oblique as well as DV views (and occasionally AP views) are often required, and would be considered superior to lateral views for diagnosis.
L269-270: CT in general (regardless whether this is CBCT, micro-CT, or helical CT) offers enhanced diagnostic capabilities over plain radiography.
L284-287: I would argue that plain radiography would also allow assessment of all these structures described. However CT would offer increased sensitivity, and allow earlier detection of osseous and dental pathology in dental disease.
L285: Do you mean the alveolar process of the maxilla and the incisor bone?
L287: intravenous radiocontrast
L 288-289: What are you referring to when describing “This technique”? I assume you are referring to contrast-enhanced CT, but I would consider CT, with or without contrast, would enable identification of concurrent disorders e.g. bulla osteitis, paranasal sinusitis/recessitis. Radiocontrast enhances soft tissue evaluation on CT.
L291-299: What are the disadvantages of CBCT when assessing soft tissues?
L269-300: The organization of this section is very confusing. Whilst the information is generally appropriate, it does not flow well and can be confusing to read. Please reorganize.
L306-307: I think this sentence is superfluous. Stomatoscopy would be the primary indication for endoscopy in rabbit dental disease, so you can start by discussing this at the beginning of this paragraph.
L313-314: Clinical crowns vs. just “crowns”. Also note previous comments regarding use of the terminology “roots”. This sentence would be better modified and expanded to provide a mini-conclusion regarding the need for a comprehensive examination including physical examination, diagnostic imaging and stomatoscopy in rabbit dental disease.
L314-327: There is a lot of repeated information here. Considering reorganizing this paragraph to make this information more succinct.
L328-332: Why is this paragraph here? Surely the ddx should be discussed earlier. Perhaps after the first paragraph in this section (6. Diagnostic approach). I’m also not sure if this was the intention of the authors, but the last sentence in this paragraph implies that a coenurus cyst cannot originate from the mandible – which is untrue.
L335-338: Are these complications associated with the presence of dental abscess or associated following treatment? I assume the latter. If so, this needs to be placed after section on treatment, otherwise this is very confusing for the reader.
L349: and the risk for ocular trauma due to exophthalmos and subsequent vision loss.
L350: The inflammation and pressure of what and from where?
L351: This sentence is poorly constructed and almost seems unnecessary. I think the key point here is to state that management of any secondary ocular disorders (which may include enucleation), must occur in conjunction with appropriate treatment of the underlying cause of retrobulbar disease.
L359-364: Here, you discussed empyema of the alveolar bulla separate to retrobulbar abscess, when in fact, they are closely related, and frequently retrobulbar abscess follows empyema of the alveolar bulla. This paragraph, together with the previous one discussing retrobulbar disease, needs reorganization.
L375: Please elaborate on how does prognosis change depending on the primary cause? As in whether this is congenital vs. traumatic vs. MBD? Is it always possible to determine the primary cause?
L382: Please define how abscesses are found “on the nose”. Do you mean within the nasal cavity or extending into the paranasal sinuses/recesses?
L385-386: I’m not sure this was the conclusion of the referenced study, given 75% of the rabbits in that study showed evidence of osteomyelitis on CT.
L386: Please define how you would classify “severe” cases e.g. multiquadrant involvement, lack of response to therapy, repeated recurrence, etc.
L398: “Surgical protocols are designed to achieve…” vs. “The aims of surgery are to achieve…”
L405-409: “Surgical removal” vs “surgical intervention”. This paragraph should be combined and summarized with the first paragraph in this section (8.1 Surgical approach) to explain why surgery is often indicated when managing rabbit dental abscesses.
L407-408: “followed by culture and sensitivity testing” vs. “based on results of microbial culture and sensitivity testing”.
L410: You have already explained why diagnostic imaging is required in the diagnosis of dental disease and for the localization of dental abscess earlier in this manuscript. So it seems unnecessary that you are recommending imaging again here. Just say that surgical intervention should be based on diagnostic imaging findings. I would also put this earlier, e.g. in the first paragraph, when discussing surgery.
L412-414: Why is this sentence here when marsupialization was discussed earlier in the second paragraph of this section?
L417-418: Why is wound packing introduced here and not earlier when discussing alternatives to complete excision of the abscess capsule? This needs to be either introduced earlier, OR delay discussing alternatives to complete excision (i.e. marsupialization and wound packing) until here.
L417-431: The key point here is that both techniques, marsupialization and wound packing, are considered alternatives when complete excision of the abscess capsule is not possible. This needs to be stated. Also, discuss the pros and cons of each technique.
L390-438: There is a lot a repetition of information in this section. This can be much better summarized.
L440-488: While all this information is valid, there is so much overlap and repetition, making it unnecessarily long and arduous to read. E.g. risk of antibiotic-inducing intestinal dysbiosis and fatal enterotoxemia was mentioned in L445/445, L446/447, L459-461, and then again in L482/483. Fluoroquinolones being “highest priority critically important antimicrobials was mentioned in L467/468, then again in L484/485.
L449-451: This has already been established in the previous section. Does not need to be repeated.
L461-463: Is this really needed here?
L464-488: Good antimicrobial stewardship is essential and important to minimize risk of antibiotic resistance. However, this does not justify four paragraphs here. Please summarize.
Figure 2: Please add reference for the studies these data were compiled from. I don’t understand how the percentages of resistance for each antibiotic were derived. How do E.coli, Klebsiella, Enterobacter, Burkholderia, and Stenotrophomonas all have the same percentages of resistance across all antibiotics? Without understanding the methodology of how these numbers were derived, the results just appear unusual.
L515-518: Does this heatmap really provide veterinarians with evidence-based insights for antibiotic selection in recurrent abscesses, cases of treatment failures, etc. as suggested? Given your statement earlier regarding “core commensals and opportunists” which show minimal resistance, should the conclusion instead be for the clinician to commence use of first-line antimicrobials as empirical antibiotic therapy to cover common bacteria in rabbit abscesses, whilst awaiting results of microbial C&S? And if involvement of less common bacteria was identified, make appropriate adjustment to the choice of antimicrobials based on sensitivity testing? I.e. as a clinician, I would not be using the table to pre-emptively treat for the unlikely possibility that Burkholdia spp. is involved, unless culture results suggest otherwise – in which case I would make my antibiotic selection based on sensitivity results, and not on Table 2.
I also think a section discussing complications and potential management strategies associated with bacterial biofilms is needed here. In these cases, recurrent or refractory infections may occur in spite of appropriate antimicrobial use.
L550: “..osteomyelitis or mandibular involvement”. I would assume these are not mutually exclusive events: if there was mandibular (bone) involvement in a dental abscess, it would suggest that osteomyelitis was present. I think just “osteomyelitis” here is sufficient.
L537-561: Please add a short (1-2 sentences) on how honey can be used topically i.e. direct application, as part of wound packing, and the frequency of use.
L602-603: This is the first time use of probiotics is discussed. What is the evidence for their use in rabbits undergoing antibiotic therapy? Consider discussing this earlier, perhaps in the section of systemic antimicrobial therapy. Also, the majority of antibiotics suitable for use in AIPMMA beads (e.g. gentamicin, ceftazidime, lincomycin, etc., not just clindamycin) would not be suitable for systemic use in rabbits due to risks of antibiotic-induced intestinal dysbiosis.
L606-607: This sentence should be placed earlier in this section – perhaps just before the third paragraph (“The preparation of AIPMMA beads poses several challenges.”)
Table 1. I’m not certain whether this table adds any significant information to the manuscript.
L619-639: There is repetition of information in this section e.g. regarding heat-stable vs heat-sensitive antibiotics, please summarize.
L563-647: This section on AIPPMA is excessively long (>3 pages). This needs to be summarized and more succinct. I would suggest rather than discussing all the antibiotics that can be incorporated in PMMA beads, select the three most commonly used ones in rabbit dental abscess management (especially since, as you have mentioned, antibiotic choice is often made pre- or intra-operatively before microbial culture results are available).
Figure 3. I think this flowchart is misleading. I am also not certain about whether it adds any value to the main text. In the majority of circumstances, microbial C&S results will indicate that both systemic and local antimicrobial treatment are indicated and appropriate. Honey can also be used in conjunction with systemic antibiotics (rather than merely an alternative to AIPMMA beads when systemic antibiotics is contraindicated, as suggested by this flowchart).
The flowchart also suggests (amongst other things), that heat-stable antibiotics should only be incorporated into PMMA beads if they are to be used in conjunction when systemic antibiotics. Obviously, this is incorrect (as only heat-stable antibiotics should be used, regardless whether it is used alone or in conjunction with systemic antibiotics), but the flowchart suggests otherwise.
As part of the post-treatment considerations, this flowchart also suggests that closure of the incision is required. Firstly, this needs to be clarified that the incision refers to the surgical site, and secondly, if honey is used, typically the surgical site is left open i.e. when marsupialization or wound packing technique is used.
L653-654: I would argue that osteomyelitis is present in most rabbit dental abscesses. I would also argue that the presence of dental abscess would be suggestive of chronic dental infection (dental abscesses rarely develop acutely) in rabbits. I am therefore not fully convinced that use of local antimicrobial therapy would be influenced by presence/absence of osteomyelitis or chronic infection. Potentially treatment of rabbit abscesses (regardless of cause) may be influenced by these factors, but not rabbit dental abscesses.
L661-676: This is a repeat of the information presented earlier in this section. This belongs in the discussion (if it is not already there), and not here.
L684-685: Not true. Lateral and DV projections are insufficient. Minimum four (to five) view skull plain radiographs are required. See Capello, 2016.
L687-688: Can you see NLD displacement on plain radiographs? I think you can assume there is displacement based on the periapical appearance of the maxillary incisors and the first maxillary cheek teeth, but to visualize the NLD, instillation of radiocontrast into the NLD (i.e. dacryocystogram) is required.
L695-696: This is the first (and only) time aspiration of an abscess is mentioned in this manuscript. Given you have mentioned earlier that surgical intervention always required, when is aspiration of an abscess for microbial C&S indicated over submission of the abscess capsule and purulent material for C&S?
L712-713: How common is hypocalcemia in pet rabbits with dental disease? While MBD and osteopenia can undoubtedly be present in pet rabbits with dental disease, this does not necessarily mean there is hypocalcemia (i.e. low serum calcium). In fact, MBD occurs in NSPHT in a bid to maintain eucalcemia.
L715-718: I would argue rather than just “routine dental checks”, a full physical examination that includes an intraoral examination is required. You mentioned earlier that many signs of dental disease in rabbits can appear extraorally, thus it would be prudent that an examination beyond the teeth would be of value.
Please also clarify what you mean by “stomatoscopic examination” – is this an intraoral exam, or are you referring to stomatoscopy – which must be performed under GA?
L719-721: “gnawing behaviour”- this primarily refers to the use of incisors. Chewable toys/branches do not hugely impact cheek teeth attrition (but forage will). I would consider using “normal masticatory pattern” instead in this instance. Note that the reference used here was a study in laboratory rabbits and looked at the use of toys to prevent stereotypical behaviors rather than the use of toys to prevent development of dental disease.
L728-730: How is this relevant as a preventative strategy/prophylaxis in “reducing the incidence and severity of odontogenic abscesses in pet rabbits (L705-706)”? Use of probiotics has no impact on dental health. The value of probiotics on the gastrointestinal health in rabbits is also equivocal based on current evidence. The study referenced here also showed the ability of probiotic E. faecium NCIMB 30183 to modify the gut flora of healthy pet rabbits, but did not investigate how this will affect rabbits being treated on antibacterials.
L750: may be highly influenced by…
L770-773: I disagree. The flowchart (i.e. Figure 3) did not offer a “tiered decision-making model”. The flowchart purely looked at local anti-microbial therapy, as suggested by its caption. A tiered decision-making model in this case should incorporate whether surgical access of the abscess is possible > if yes, then whether complete surgical excision of the abscess capsule is possible (and if not, then…) > if yes, then excise > if not, then consider marsupialization or wound packing technique > submit section of abscess capsule and purulent material for C&S > commence empirical systemic antibiotics (and offer possible appropriate choices) while awaiting results + local antimicrobial therapy (choices) > modify antimicrobial choices +/- modality based on C&S results
L771: There is no mention of surgery at all in the flowchart except for, “ensure secure closure of incision”.
L790: A section (around one paragraph) is needed here to discuss strategies and prognosis when surgery (complete excision, or marsupialization, or wound packing) on the abscess is not possible due to location e.g. in the retrobulbar space
Author Response
Dear reviewer,
Thank you for your constructive feedback; we have revised the manuscript accordingly and provide a point-by-point response below, detailing updates to content, focus, and visual presentation.
We provide as an attached file, a detailed, point-by-point response to each comment, outlining the changes that have been made.

Reviewer 3 Report
Comments and Suggestions for Authors
Introduction
The purpose of this paper is not detailed, nor is there any reference to the existence or absence of similar studies. See recommended bibliography for examples.
- Etiology and Pathogenesis
The difference between the different sexes is not mentioned. There is literature on this subject; it should be noted in the paper.
Examples: In the United Kingdom, it was observed that neutered rabbits were 1.38 times more likely to suffer from dental disease than intact rabbits. This observation may be due to a bias in clinical care rather than an actual link between neutering and dental disease, as owners of neutered rabbits may have a stronger human-animal bond and therefore be more likely to present their rabbit for regular veterinary examinations when they detect early signs of dental disease. Furthermore, male rabbits were 1.23 times more likely to suffer from dental disease than female rabbits in this study. Males had a statistically significantly higher prevalence of overgrown nails than females.
In Chile, age and male sex were found to be significant risk factors for acquired dental disease. In contrast, a free-roaming lifestyle and dietary hay intake were protective factors. This study, along with the one from the United Kingdom, supports the predisposition of male rabbits to dental disease.
References:
- Conway RE, Burton M, Mama K, Rao S, Kendall LV, Desmarchelier M, Sadar MJ. 2023. Behavioral and Physiologic Effects of a Single Dose of Oral Gabapentin in Rabbits (Oryctolagus cuniculus). Top Companion Anim Med., Mar-Jun; pp 53-54: 100779.
2.Jackson MA, Burn CC, Hedley J, Brodbelt DC, O'Neill DG. 2024. Enfermedad dental en conejos de compañía bajo atención veterinaria primaria del Reino Unido: frecuencia y factores de riesgo. Rec. Veterinario; E3993.
3. Mosallanejad B, Moarrabi A, Avizeh R, Ghadiri A. 2010. «Prevalencia de maloclusión dental y elongación de la raíz en conejos domésticos de Ahvaz, Irán», Revista Iraní de Ciencia y Tecnología Veterinaria, 2(2), pp. 109-116.
4.O'Neill DG, Craven HC, Brodbelt DC, Church DB, Hedley J. 2020. Morbilidad y mortalidad de conejos domésticos (Oryctolagus cuniculus) bajo atención veterinaria primaria en Inglaterra. Recreación veterinaria; 186:451.
5. Palma-Medel T, Marcone D, Alegría-Morán R. 2023. Enfermedad dental en conejos (Oryctolagus cuniculus) y sus factores de riesgo: un estudio de práctica privada en la Región Metropolitana de Chile. Animales, 13, 676.
6. Tokashiki E, Rahal S, Melchert A, Gonçalves R, Rolim L, Teixeira C. 2018. Estudio retrospectivo de las condiciones agrupadas por sistemas corporales en conejos de compañía. Revista de Medicina de Mascotas Exóticas. 29. 10.1053/j.jepm.2018.11.009.
Author Response
Dear reviewer,
We would like to express our sincere thanks for the thoughtful and comprehensive evaluation of our manuscript. We have carefully reviewed all general and specific comments, and we greatly appreciate the clarity and depth of the feedback provided. These observations have helped us identify areas where the manuscript’s structure, coherence, and focus can be improved. In response, we have undertaken several revisions to enhance the organization and readability of the review.
We have enclosed the detailed point by point responses.

Round 2
Reviewer 3 Report
Comments and Suggestions for Authors
Good afternoon. The paper is fine.
Author Response
Thank you for the positive feedback.